# From Supervision to Exploration: What Does Protein Language Model Learn During Reinforcement Learning?

## Abstract

Protein Language Models (PLMs) have achieved significant breakthroughs in computational protein science through pre-training on large-scale sequence databases and leveraging scalable network architectures. Concurrently, Reinforcement Learning (RL) has demonstrated substantial progress across multiple protein design tasks by enabling expanded exploration capabilities and precise multi-objective optimization. While RL has shown transformative potential in natural language processing by enabling models to discover emergent capabilities beyond their training distributions, its capacity to unlock latent functional patterns within protein sequence space remains underexplored. In this study, we investigate whether RL-enhanced PLMs can transcend their pre-training limitations and identify implicit sequence-structure-function relationships not explicitly encoded in foundational datasets. Through systematic evaluation across four critical protein design domains—*antimicrobial peptide (AMP) design*, *kinase optimization*, *antibody engineering*, and *inverse folding*—we employ diverse RL algorithms and model architectures to address this fundamental question. Our comprehensive analysis demonstrates that RL reliably improves sampling efficiency across domains and, more importantly, that its effectiveness is governed by a three-factor interaction: *task difficulty*, *reward model accuracy*, and *policy capacity*. Gains scale when rewards are accurate and informative, policies have sufficient capacity to realize the signal, and tasks present headroom beyond supervised learning; conversely, noisy rewards or capacity bottlenecks cap improvements despite exploration. This principled view offers practical guidance for RL in protein design: prioritize reward refinement before scaling policy size, match RL algorithms and regularization strength to task difficulty, and allocate capacity where marginal gains are largest.

## 1 Introduction

Protein Language Models (PLMs) have emerged as the cornerstone of computational protein design, leveraging vast training datasets and scalable network architectures to achieve remarkable success across feature representation (Lin et al., 2023; Hayes et al., 2024; Brandes et al., 2022), sequence generation (Nijkamp et al., 2023; Ferruz et al., 2022; Bhatnagar et al., 2025; Truong Jr & Bepler, 2023), and functional prediction (Su et al., 2023; Hayes et al., 2024; Xu et al., 2023a). These advances have successfully propelled the development of sequence-function relationship studies and protein design applications (Qiu et al., 2024; Zhang et al., 2025; Ruffolo et al., 2025).

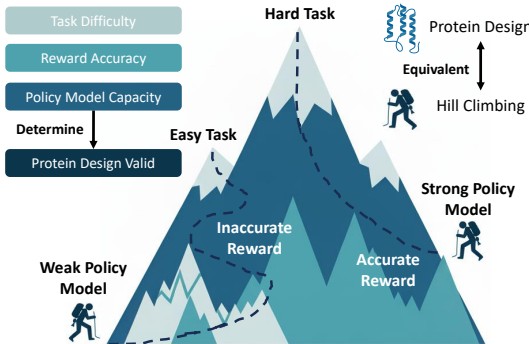

Figure 1: Reinforcement learning for protein design is akin to hill climbing. Task difficulty equates to mountain height, policy model capacity to the starting altitude, and reward accuracy to direction correctness. These three factors jointly determine the RL efficacy in protein design.

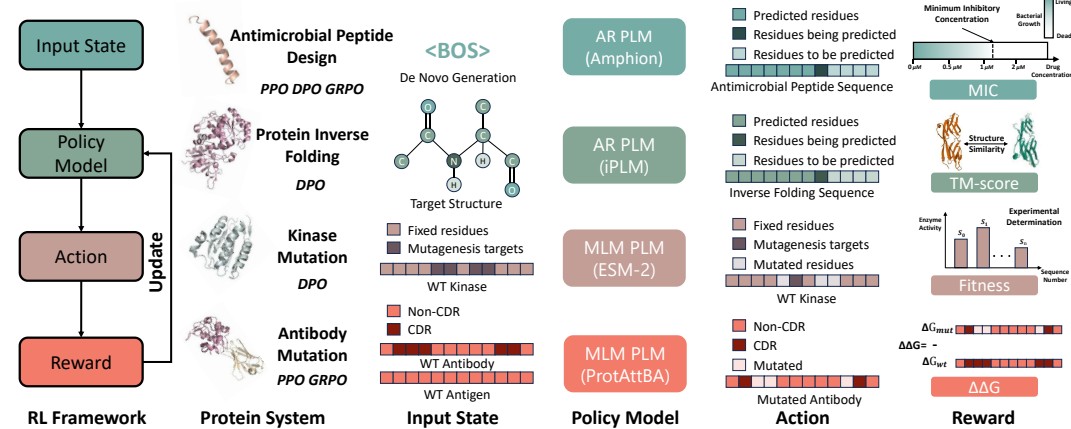

Figure 2: **Overview of the four biological systems** based on PLM and RL. **AR and MLM** denote Auto-regressive and Masked Language Modeling, respectively.

However, functional protein design reveals fundamental limitations of supervised learning approaches. Traditional methods face three critical obstacles: first, the inability to optimize for complex, non-differentiable biological objectives such as TM-Score (Zhang & Skolnick, 2005) that often require iterative refinement (Yang et al., 2019); second, being constrained to interpolate within existing sequence-function mappings, thereby struggling to explore novel functional regions (Johnston et al., 2023; Notin et al., 2023); third, the inability to integrate multi-objective criteria or real-time experimental feedback (Jiang et al., 2024; Yang et al., 2025). These limitations restrict the discovery of innovative protein sequences, creating a critical gap between computational capabilities and practical engineering requirements.

Reinforcement Learning directly addresses these challenges by enabling exploration beyond observed data, supporting multi-objective optimization, and integrating expert or experimental feedback at scale. Recent studies, which are summarized in Table 4, have demonstrated the transformative potential of RL across multiple protein design tasks (Lutz et al., 2023; Xu et al., 2025; Wang et al.). When coupled with PLMs, RL gains additional power. Numerous current studies have substantiated this advantage. For instance, EvoPlay (Wang et al., 2023) discovered fluorescent proteins with several-fold higher activity than wild-type through Monte Carlo tree search exploration. ProteinZero (Wang et al., 2025b) developed proteins with enhanced designability, thermostability, and greater diversity through diversity-based Generalized Reward-based Policy Optimization (GRPO) (Shao et al., 2024). ApexAmphion (Cao et al., 2025b) successfully explored broader and more potent AMP candidates through Proximal Policy Optimization (PPO) (Schulman et al., 2017). These methods transcend the limitations of supervised training through reward-based exploration.

Simultaneously, developments in natural language processing have revealed RL's potential for enhancing task performance and developing novel reasoning strategies (Liu et al., 2025c), though some research suggests that RL primarily amplifies existing outputs (Yue et al., 2025; Wu et al., 2025). This raises a fundamental question:

*Do new emergent capabilities arise during the RL fine-tuning process of PLMs?*

To the best of our knowledge, this study is the first to systematically evaluate this question in the context of protein design. We conduct experiments across four biological systems—*antimicrobial peptide design*, *kinase optimization*, *antibody mutation*, and *protein inverse folding*—to probe how RL interacts with PLMs. Our results show that RL consistently improves sampling efficiency for beneficial sequences. More importantly, we find that RL's effectiveness is determined by the interaction of three key factors: *task difficulty*, defined by the ruggedness and observability of the underlying fitness landscape; *reward model accuracy*, reflecting how well the reward signal is calibrated and how much signal-to-noise it conveys; and *policy model capacity*, which depends on model size, representational power, and initialization quality. As shown in Fig. 1, RL training for protein design can be likened to hill-climbing: task difficulty sets the height of the summit to be scaled, reward

accuracy determines the climbing direction, and policy-model capacity fixes the starting altitude. These factors jointly shape whether RL can climb towards subspaces with stronger task alignment or stall in suboptimal plateaus. Different combinations of task complexity, reward fidelity, and policy strength yield qualitatively distinct trajectories of improvement. We believe this framework provides a principled way to measure current RL–PLM systems and serves as a practical blueprint for guiding future RL applications in protein design.

## 2 METHOD

**Notations.** We define a unified framework for protein sequence optimization tasks. Let $\mathcal{A} = \{A, C, D, E, F, G, H, I, K, L, M, N, P, Q, R, S, T, V, W, Y\}$ denote the set of 20 natural amino acids, which compose the vocabulary of protein design, and $\mathcal{S} = \mathcal{A}^*$ represent the space of all finite protein sequences.

### 2.1 PROTEIN INVERSE FOLDING

For protein inverse folding, we address structure-to-sequence mapping where the policy model $\pi_\theta(\mathbf{s}|\mathbf{z})$ generates sequences conditioned on target 3D structure $\mathbf{z} \in \mathcal{Z}$ (Xu et al., 2025). The optimization objective combines sequence likelihood with designability constraints:

$$\mathbf{s}^* = \arg \max_{\mathbf{s} \in \mathcal{S}} \mathbb{E}_{\mathbf{s} \sim \pi_\theta(\cdot|\mathbf{z})}[\log p(\mathbf{s}|\mathbf{z}) + \lambda \Xi(\mathbf{s}, \mathbf{z})], \tag{1}$$

where $p(\mathbf{s}|\mathbf{z})$ represents the structure-conditioned sequence probability and $\Xi(\mathbf{s}, \mathbf{z})$ captures designability constraints that ensure the generated sequence can fold into the target structure.

We employ InstructPLM-7B (Qiu et al., 2024) as our policy model, initially trained on the CATH 4.2 dataset (Sillitoe et al., 2021) to establish inverse folding capabilities. The action space corresponds to autoregressive sequence generation, where at each step $t$, the policy selects amino acid token $a_t \in \mathcal{A}$ according to:

$$a_t \sim \pi_\theta(a_t|\mathbf{z}, a_{1:t-1}), \tag{2}$$

where $\mathbf{z}$ represents the target structure and $a_{1:t-1}$ denotes previously generated tokens. The complete sequence is constructed through iterative token selection until reaching the end-of-sequence token.

We applied TM-Score as the reward function for structural fidelity evaluation. By employing ESM-Fold (Lin et al., 2023) to predict the structure $\mathbf{z}_{pred}$ for each generated sequence, we calculate the TM-Score as TM-Align($z$, $z_{pred}$) (Zhang & Skolnick, 2005).

We then implement Direct Preference Optimization (DPO) (Ferruz et al., 2024) enhanced with regularization. For each target structure, we sample sequences using the current policy model, evaluate the TM-Scores, and rank them to create preference pairs with high-scoring sequences as positive examples $S_w$ and low-scoring sequences as negative examples $S_l$. The loss function combines standard DPO with supervised regularization (Xue et al., 2025):

$$\mathcal{L}(\pi_\theta, \pi_{\text{ref}}) = -\mathbb{E}_{(\mathbf{z}, S_w, S_l) \sim D_{pair}} \Big[ \underbrace{\log \sigma \Big( \beta \log \frac{\pi_\theta(S_w|\mathbf{z})}{\pi_{\text{ref}}(S_w|\mathbf{z})} - \beta \log \frac{\pi_\theta(S_l|\mathbf{z})}{\pi_{\text{ref}}(S_l|\mathbf{z})} \Big)}_{\mathcal{L}_{\text{DPO}}} - \underbrace{\lambda \log(\pi_\theta(S_w|\mathbf{z}))}_{\mathcal{L}_{reg}} \Big], \tag{3}$$

where $\mathcal{L}_{reg}$ maintains sequence fidelity by encouraging the model to assign high probability to structurally superior sequences. We employ multi-round iterative refinement, where each round generates updated preference data and refreshes reference weights for progressive improvement.

### 2.2 ANTIMICROBIAL PEPTIDE DESIGN

For AMP design, we generate peptides with enhanced antimicrobial activity by targeting lower MIC (minimum inhibitory concentration) values, indicating stronger bacterial inhibition. We employ Amphion-SFT (Cao et al., 2025b), an autoregressive PLM trained on AMPs, as our policy model $\pi_\theta(\mathbf{s})$ to generate sequences optimized for antimicrobial potency. The optimization objective is:

$$\mathbf{s}^* = \arg \max_{\mathbf{s} \in \mathcal{S}_{\text{AMP}}} \mathbb{E}_{\mathbf{s} \sim \pi_\theta}[f_{\text{MIC}}(\mathbf{s})], \tag{4}$$

where $f_{\text{MIC}} : \mathcal{S}_{\text{AMP}} \to \mathbb{R}$ denotes ApexMIC (Cao et al., 2025b), a binary classifier for predicting antimicrobial potential. The reward function transforms the predicted score through normalization:

$$R(\mathbf{s}) = 2 \cdot (f_{\text{MIC}}(\mathbf{s}) - \lambda), \tag{5}$$

where $\lambda = 0.4$ denotes the threshold for binary classification, providing balanced reward estimation.

We implement DPO (Ferruz et al., 2024), PPO (Schulman et al., 2017), and GRPO (Shao et al., 2024) for fine-tuning. An additional KL regularization term is added in the loss of PPO and GRPO to keep the naturalness of generated AMPs. Detailed formulations are shown in Appendix C.4.

## 2.3 KINASE MUTATION

The kinase mutation task requires the model to perform multi-step mutations at specified positions of the initial sequence, where each step involves selecting both the mutation site and the amino acid substitution to progressively enhance the final mutant's fitness. Consider a wild-type protein sequence $\mathbf{s}_0 = (s_{0,1}, s_{0,2}, \ldots, s_{0,n}) \in \mathcal{S}$ where $s_{0,i} \in \mathcal{A}$. The fitness optimization objective is:

$$\mathbf{s}^* = \arg\max_{\mathbf{s}' \in S} \Phi(\mathbf{s}'), \tag{6}$$

where $\Phi : \mathcal{S} \to \mathbb{R}$ quantifies protein fitness.

We adopt the ESM-2 architecture as our base model and follow the training framework described in Wang et al. (2024). The annotated fitness value serves as the reward, and the policy model performs multi-step mutations on the wild-type sequence $s_0$ to maximize fitness of the final sequence $s_t$. The action space consists of position selection and amino-acid selection defined as $a_t = (\hat{p}_t, \hat{x}_t)$ where $\hat{x}_t \neq s_{t-1}[\hat{p}_t]$. The policy model uses ESM2 embeddings and MLP to predict mutation position, then replaces the corresponding residue with [MASK] token and employs ESM-2 to select the new amino acid. During DPO training, we did not employ either the KL penalty or the entropy regularization term (See details in Section C.4).

## 2.4 ANTIBODY OPTIMIZATION

For antibody optimization, we consider the complex space $\mathcal{C} = \mathcal{S}_{ab} \times \mathcal{S}_{ag}$ where $\mathcal{S}_{ab}$ and $\mathcal{S}_{ag}$ represent antibody and antigen sequence spaces. Given a fixed antigen sequence $\mathbf{s}_{ag}$, the policy model $p_\theta(\mathbf{s}_{ab}|\mathbf{s}_{ag})$ aims to generate optimized antibody sequences that minimize binding affinity change:

$$\mathbf{s}_{ab}^* = \arg\min_{\mathbf{s}_{ab}} \mathbb{E}_{\mathbf{s}_{ab} \sim p_\theta(\cdot|\mathbf{s}_{ag})}[\Delta\Delta G(\mathbf{s}_{ab}, \mathbf{s}_{ag})], \tag{7}$$

where $\Delta\Delta G(s_{ab}, s_{ag}) = \Psi(\mathbf{s}_{ab}, \mathbf{s}_{ag}) - \Psi(\mathbf{s}_{ab}^{wt}, \mathbf{s}_{ag})$ denotes the binding affinity change from wild-type, which is predicted by a re-implemented version of ProtAttBA (Liu et al., 2025a). The new architecture is designed to better model the action of the policy model through logits (See details in Alg.1). Instead of training by regression on $\Delta\Delta G$ (Liu et al., 2025a), we re-designed its training loss with combined objectives:

$$\mathcal{L}_{total} = \mathcal{L}_{reg} + \lambda \mathcal{L}_{MLM}, \tag{8}$$

where $\mathcal{L}_{reg}$ represents the original $\Delta\Delta G$ regression loss and $\mathcal{L}_{MLM}$ denotes masked language modeling (MLM) loss. We achieved higher performance on the test set (Tab. D.1).

The wild-type antibody sequence is mutated through policy logits $\mathbf{z} \in \mathbb{R}^{L \times |\mathcal{A}|}$ where $L$ denotes sequence length, combined with a position head that selects mutation sites within CDR. Mutated sequences are generated through multinomial sampling from the policy logits of antibody at selected CDRs, with detailed procedures described in Algorithm 2. We employ rollout mechanisms and compute Generalized Advantage Estimation (GAE), with a value model initialized using an MLP architecture.

We applied both PPO and GRPO for model training. The comprehensive loss function is as follows:

$$\mathcal{L}_{total} = \mathcal{L}_{policy} + \alpha \mathcal{L}_{KL} + \beta \mathcal{L}_{value} + \gamma \mathcal{L}_{entropy}, \tag{9}$$

where $\mathcal{L}_{policy}$ denotes the standard policy loss for PPO and GRPO (See details in Appendix C.4), $\mathcal{L}_{KL}$ denotes the clamped KL divergence loss between current and reference policies, $\mathcal{L}_{value}$ denotes the regression loss for value function, and $\mathcal{L}_{entropy}$ represents position entropy computed over mutation position logits to encourage exploration. The coefficients $\alpha$, $\beta$, and $\gamma$ balance the contribution of each component.

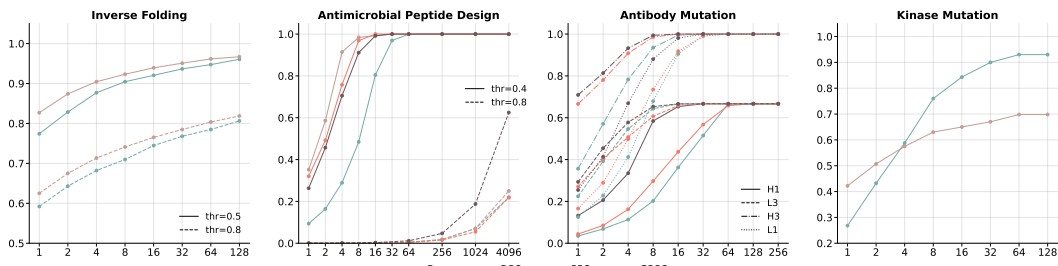

Figure 3: Pass@k results for four biological systems.

# 3 EXPERIMENTS

## 3.1 DATASETS AND EVALUATION METRICS

**Datasets** The kinase mutation experiments utilize PhoQ (Podgornaia & Laub, 2015) containing 140,517 annotated variants from 160,000 ($20^4$) possible mutations at four sites (A284, V285, S288, T289), with unlabeled variants assigned fitness values of -1. The antibody mutation task employs the AB1101 dataset (Wang et al., 2020) comprising 32 antigen-antibody complexes with 645 single-point mutations for training and 456 multi-point mutations for testing, where complexes 1MLC and 1VFB serve as designated test structures. RL design leverages sequences from DBAASP, DRAMP, and APD3 databases Pirtskhalava et al. (2021); Shi et al. (2022); Wang et al. (2016), yielding 7,888 samples (peptides 6-50 amino acids, active threshold <32 $\mu$M/mL MIC) split via MMseqs2 (Steinegger & Söding, 2017) clustering into 6,153 training, 789 validation, and 946 test samples. Protein inverse folding experiments use CATH4.2 (Sillitoe et al., 2021) with 18,024 training structures for base model and DPO training, evaluated on the CATH4.2 test set (1,120 structures) combined with TS50 (50 structures) and TS500 (470 structures) benchmarks.

**Evaluation Metric** **Pass@k** metric is applied to evaluate model's sampling efficiency for objective-satisfying feasible sequences, which is calculated by the probability of succeeding at least once by taking the complement of the probability of failing in all $k$ consecutive trials. It is formally defined as:

$$\text{Pass@}k(p) = 1 - \left(\text{Pr}_{y \sim p(y|x)}[R(x, y) = 0]\right)^k, \quad (10)$$

where $p(y|x)$ is the output probability distribution of the model for a given prompt $x$. $R(x, y)$ is a reward function that returns 1 if the completion $y$ is correct, and 0 otherwise. The term $\text{Pr}_{y \sim p(y|x)}[R(x, y) = 0]$ denotes the probability of an incorrect answer in a single sample.

To analyze the deviation of pre- and post-RL models, we follow (Wu et al., 2025)'s definition of **Support** as a more detailed metric for pass@k. We leverage three key concepts under pass@k setting. *Shrinkage*(k) represents the set of problems that the base model could solve but the fine-tuned model cannot solve at pass@k. *Expansion*(k) denotes the set of problems that the base model could not solve but the fine-tuned model can now solve at pass@k. *Preservation*(k) refers to the set of problems that both models can solve at pass@k. To quantify the trade-off between discovering new correct solutions and forgetting previously known ones, we propose the *Expansion-Shrinkage Ratio (ESR)*, a simple yet effective metric that captures the balance between knowledge gain and loss during fine-tuning:

$$\text{ESR}(k) = |\text{Expansion}(k)|/|\text{Shrinkage}(k)|. \quad (11)$$

An ESR greater than 1.0 indicates net knowledge gain, while an ESR less than 1.0 signals net knowledge loss, and an ESR equal to 1.0 represents balanced learning dynamics.

**Biological Metrics.** We applied multiple metrics to evaluate the biological reasonability. (i) *Positional entropy* is applied to evaluate sequence diversity at individual positions within the complementarity-determining regions of both kinase and antibody mutation tasks, measuring the uncertainty across mutational sites. (ii) *Perplexity* is applied to evaluate the likelihood quality of generated protein sequences across all tasks, computed as the exponential of the average negative

Table 1: Results for *Support* metric for four biological systems.

|  | Preservation | Expansion | Shrinkage | Out-of-support | ESR ↑ |
|---|---|---|---|---|---|
| AMP design | 290 | 7 | 49 | 4 | 0.14 |
| Kinase mutation | 260 | 8 | 100 | 32 | 0.08 |
| Antibody mutation | 8 | 2 | 4 | 2 | 0.50 |
| Inverse folding | 891 | 9 | 21 | 199 | 2.33 |

log-likelihood under the respective models. (iii) *Diversity and Novelty* are applied to evaluate sequence variation and distinctiveness within generated outputs for both RL design and protein inverse folding tasks, calculated as average sequence similarity between generated sequences and the average of one minus maximum sequence similarity scores, respectively. (iv) *Recovery rate* and *TM-score* are applied to evaluate sequence similarity and structure similarity in protein inverse folding tasks.

## 3.2 INVERSE FOLDING

In the inverse folding task, RL fine-tuned the base model, resulting in higher TM-scores, as demonstrated by the improved pass@k performance and TM-score distribution (Figure 4C). Across all values of k and both thresholds (0.5 and 0.8), the RL model consistently outperformed the base model, indicating more effective exploration toward higher-quality structural predictions. This suggests that RL learned to focus on sequence–structure relationships that maximize TM-scores.

The RL model showed lower perplexity in the DPO variant (Figure 4B), indicating that it sampled more efficiently compared to the base model. However, as shown in Figure 4A, the RL model exhibited slightly reduced novelty and diversity but higher recovery, suggesting that RL exploration prioritized regions with high TM-scores, at the cost of reduced exploration in diverse regions.

When stricter evaluation criteria were applied (TM-score > 0.8 and sequence similarity < 0.7), expansion cases outnumbered shrinkage cases across all k-values, demonstrating that RL exploration expanded the design space (Figure 7). Smaller k-values, however, resulted in decreased ESR, suggesting that RL's sampling efficiency is more pronounced at larger k-values.

UMAP visualization (Figure 4D) further supports this conclusion, showing that the RL model's sampling distribution aligns with a subset of the base model's, with distinct expansion and shrinkage regions. This indicates that RL's exploration is focused on high-quality structural solutions while maintaining a core subset of diverse sequences.

In the inverse folding task, RL fine-tuned the base model, effectively navigating the task's moderate complexity and rugged landscape. This allowed the model to prioritize high-quality structural solutions, as indicated by the improved TM-scores and pass@k performance (Figure 4C). The RL model's reduced perplexity (Figure 4B) suggests more efficient sampling, with exploration directed toward high-reward regions characterized by better structural similarity. However, this focus on high-reward regions came at the cost of diversity and novelty, as seen in the slightly reduced values for these metrics (Figure 4A).

## 3.3 ANTIMICROBIAL PEPTIDE DESIGN

In the AMP design task, RL fine-tuned the base model to generate AMPs with lower MIC values. As shown in Figure 4F, the RL model outperformed the base model, with approximately 95% of the generated samples achieving lower perplexity, indicating more efficient sampling. This suggests that RL effectively directed exploration toward high-reward regions associated with lower MIC values.

In the pass@k evaluation (Figure 3), all RL models (DPO, PPO, and GRPO) outperformed the base model, particularly under the challenging 0.8 threshold for binary classification. While DPO and PPO performed similarly to the base model, GRPO showed continuous improvement, reflecting its superior exploration ability. This performance boost can be attributed to GRPO's group loss mechanism, which emphasizes high-reward samples and encourages more focused exploration.

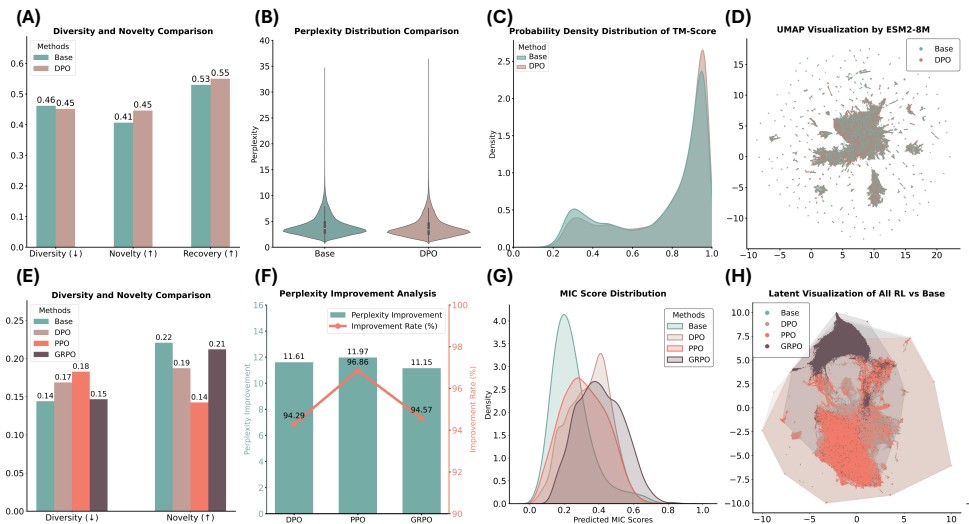

Figure 4: Experimental results for inverse folding (A-D) and AMP design (E-H).

Next, we assessed sampling efficiency by constructing support sets with a cross-entropy threshold < 3.0 (Figure 4G). RL methods discarded more positive samples (ESR = 0.14) but achieved higher sampling efficiency, demonstrating a more targeted exploration approach. UMAP visualization of latent distributions further confirmed these findings: RL models, especially GRPO, concentrated within a high-reward subset of the AMP space. GRPO's distribution showed a clear shift within the base model's convex hull, indicating that it learned to focus on promising regions while maintaining coverage across the original design space. In contrast, DPO and PPO models showed reduced diversity and novelty, emphasizing GRPO's superior exploration capability.

The AMP design task involves navigating a rugged and complex search space, where low MIC values are rare (Figure 3B). RL successfully focused exploration on regions associated with lower MIC values, demonstrating its ability to tackle challenging tasks with a well-structured reward signal. The reward model's accuracy, as reflected in Table F.1, shows that RL effectively exploited the reward signal. GRPO, in particular, highlighted the significance of policy model capacity, outperforming DPO and PPO by achieving stronger exploration and prioritizing high-reward regions while maintaining coverage within the original search space.

## 3.4 KINASE MUTATION

In the kinase mutation task, RL fine-tuned the base model by prioritizing high-fitness sequences, at the cost of overall sequence diversity. This shift reflects the optimization objective's focus on maximizing fitness within a rugged, discontinuous protein landscape. Sampling 50,000 sequences revealed minimal overlap (76 sequences, 4%) between the base and RL models. Notably, entropy at mutable positions shifted: the RL model showed a decrease at position 1 and marked increases at positions 2 and 3, indicating a fundamental distributional shift (Figure 5A-B).

The RL model achieved a significantly higher mean fitness, with peak scores reaching 133, compared to 70 for the base model. However, the base model maintained superiority in low-fitness regions (<1), as shown in Figure 5C. Pass@k evaluation (Figure 3) confirmed this trend: RL excelled at k=1-2 but was overtaken by k=4, with the base model leading at saturation (k=128). When we assessed support sets at k=32, starting with 400 wild-type sequences from the test set, the results revealed that RL led to a contraction of the sequence space, with an ESR of only 0.08. This indicates that RL training in the kinase mutation task sacrifices some exploration capacity to focus on high-fitness regions, leading to better performance within these areas.

UMAP visualization (Figure 5D) further corroborates this finding. The RL model's sampling distribution was more concentrated compared to the base model, indicating a shift in probability mass toward high-fitness regions. While most RL sequences formed a subset of the base model's distribu-

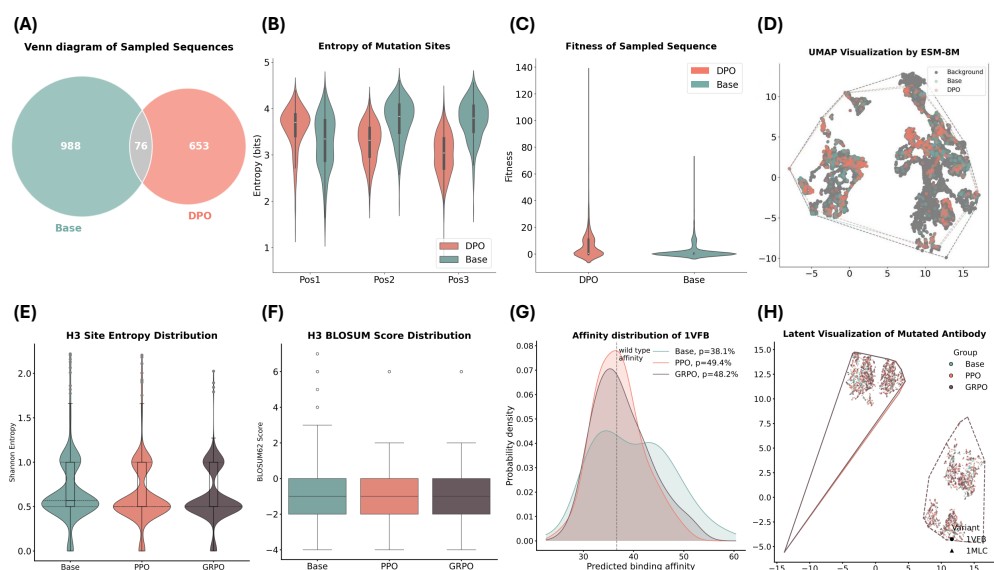

Figure 5: Experimental results for kinase mutation (A-D) and antibody optimization (E-H).

tion, a few escaped the original convex hull, suggesting some degree of novel exploration. However, the overall RL distribution largely resembled a subset of the base model, with limited exploration outside of the original distribution.

Overall, RL effectively optimized for high-fitness regions but faced challenges in balancing exploration and exploitation due to the task's complex and discontinuous fitness landscape. The verified reward model enabled RL to focus on high-reward regions, as reflected in higher mean fitness and lower perplexity, directing sampling towards high-fitness areas. However, due to a weaker policy model (as seen in pass@k), RL's exploration efficiency was limited (ESR = 0.08). While the RL model outperformed in high-fitness regions, its limited diversity indicates that the task requires a stronger policy model to further enhance exploration.

### 3.5 ANTIBODY MUTATION

In the antibody mutation task, RL models fine-tuned on different CDRs (L1, L3, H1, H3) generally achieved higher pass@k values, with GRPO outperforming PPO (Figure 3). Pass@k reached 1.0 for H3 and L1 sites, while H1 and L3 tasks proved more challenging, with convergence to 0.67. These results suggest that RL effectively optimized the sampling in more accessible regions but faced difficulties in challenging tasks. Support evaluation on the test set, based on the average cross-entropy of L3 CDR sites, confirmed that RL models demonstrated higher sampling efficiency (Figure 5G). However, in some cases, shrinkage exceeded expansion (ESR = 0.5), indicating net knowledge loss during RL training. This suggests that while RL can improve efficiency, it may also limit diversity when the exploration is overly concentrated on specific regions (Table 3.1).

Further analysis of mutated sites revealed that RL models, particularly PPO and GRPO, generated mutations with lower entropy values, while maintaining BLOSUM substitution score distributions similar to the base model (Figure 5E-F). This suggests that RL models learned to prefer more conservative mutation strategies, adhering to physicochemical constraints while optimizing for specific amino acid substitutions.

In terms of the reward function, RL models shifted towards lower ddG values, indicating a preference for more favorable energy states. The distribution of ddG values showed that RL explored regions with significantly lower ddG (Figure 6). To validate these findings and exclude potential reward-hacking artifacts, we compared the results with Protenix-Mini (Gong et al., 2025) for structural prediction of 1VFB H3 variants and FoldX (Schymkowitz et al., 2005) for affinity prediction (Figure 5G). RL models showed consistent improvements on this independent affinity function, with more concentrated distributions, despite a reduced proportion of low-energy samples.

Furthermore, analysis of reward function ddG distributions showed that RL models shifted toward lower ddG values and explored regions with substantially lower ddG (Figure 6). To exclude reward-hacking artifacts from the imperfect reward model (Spearman correlation = 0.47), we validated results using Protenix-Mini (Gong et al., 2025) for structural prediction of 1VFB H3 variants and FoldX (Schymkowitz et al., 2005) for affinity prediction (Figure 5G). RL models demonstrated consistent improvements on this independent affinity function, with more concentrated distributions despite reduced low-energy sample proportions. Visualization of two test set PDB variants showed no significant convex hull boundary shifts after RL (Figure 5H).

The antibody-antigen mutation task presents a more complex landscape, with certain CDR regions being more easily optimized than others. RL effectively optimized these more accessible regions, but the limited success in H1 and L3 tasks highlights challenges in exploring more difficult areas. These challenges stem from the relatively low accuracy of the reward model (Spearman = 0.47) and suboptimal policy model initialization (as evidenced by pass@k results). While RL prioritized exploration of energetically favorable regions, this also led to a trade-off in diversity and shrinkage (ESR = 0.5). Compared to other tasks, the antibody domain presents significantly greater challenges, requiring more valuable work to enhance the model's exploration capability and generalizability.

## 4 RELATED WORKS

Current RL approaches for protein design focus on five major tasks. For **structural design** tasks, MCTS-based approaches dominate due to their ability to handle complex architectural constraints. Lutz et al. (Lutz et al., 2023) and GAPN (Gao et al., 2024) designed multimer protein complex assembly with MCTS and PPO, respectively. **Sequence optimization** represents the most diverse application area, employing various algorithms depending on multiple optimization objectives. MCTS-based methods include EvoPlay (Wang et al., 2023) for enzyme design and RelaVDEP (Mi et al., 2025) for multi-objective protein engineering. PPO-based approaches encompass RLXF (Blalock et al., 2025) for experimental feedback integration, $\mu$Protein (Sun et al., 2025) for landscape model-guided design, and ApexAmphion (Cao et al., 2025b) for antimicrobial peptide optimization. For **inverse folding**, recent work has gravitated toward DPO and its variants, including multi-round DPO (Xu et al., 2025), EnerBridge-DPO (Rong et al., 2025), and ResiDPO (Xue et al., 2025). Alternative approaches include ProteinZero using PPO/GRPO (Wang et al., 2025b) and Prot-InvTree employing MCTS (Liu et al., 2025b). **Antibody engineering** applications utilize diverse RL paradigms: AB-Gen (Xu et al., 2023b) with REINVENT for CDRH3 libraries, BetterBodies (Vogt et al., 2024) and structured Q-learning (Cowen-Rivers et al., 2022) for Q-learning-based optimization, and stability-focused approaches using reward fine-tuning (Wang et al.) and PPO (Cao et al., 2025a). Finally, **peptide binder design** employs specialized algorithms: TCRPPO (Chen et al., 2023) for T-cell receptor sequences, and MCTS-based methods like HighPlay (Lin et al., 2025) and CYC_BUILDER (Wang et al., 2025a) for cyclic peptide optimization.

## 5 CONCLUSION

This study is the first to directly explore what reinforcement learning (RL) can teach protein language models (PLMs) in protein design tasks. Through an analysis of two leading PLM architectures, three RL algorithms, and four prominent experimental systems, we conclude that RL enables more efficient sampling of high-reward regions. However, RL's ability to learn new patterns and optimize high-reward distributions comes with trade-offs, including reduced diversity, increased shrinkage (ESR < 1), and other costs. These effects are influenced by factors such as the initialization capacity of the policy model (base model), reward accuracy, and task complexity. We believe this insight offers a meaningful explanation for the current landscape of RL-based protein design. Building on this, researchers can adopt a fresh perspective on how to make RL fine-tuning more effective. While this work primarily focuses on PLMs for protein sequence design, future research will extend to Diffusion/Flow Matching architectures, protein structure and sequence-structure co-design, and additional RL algorithms (e.g., MCTS). We anticipate that the findings from this study, coupled with future validation, will provide valuable insights that drive innovation in the field.

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

## A  IMPLEMENTATION DETAILS

### A.1  PROTEIN INVERSE FOLDING

**Training Details**  Following (Xu et al., 2025), the DPO framework used $\beta = 0.5$ to balance reference model retention with preference adaptation, and regularization weight $\lambda = 1$ for optimal learning from chosen and rejected sequence pairs. Training employed AdamW optimizer with learning rate $1 \times 10^{-5}$, $\beta_1 = 0.9$, $\beta_2 = 0.999$, $\epsilon = 1 \times 10^{-8}$, and batch size 128 across 8 NVIDIA A100 GPUs. LoRA adaptation used rank $r = 16$ and $\alpha = 16$, training only 0.1% of total parameters.

Single-round training used 4,000 steps with 20 sequences per structure, while multi-round training employed 200 steps per round across 20 rounds using 200 sequences per structure. The reference model $\pi_{ref}$ was reinitialized with previous iteration weights during multi-round training to maintain preference alignment and prevent catastrophic forgetting.

**Sampling Details**  Sequence generation employed distinct sampling parameters for training and evaluation phases. For **Training**, single-round training used temperature $T = 1.0$ and top-p = 0.9 to generate 20 sequences per structure, while multi-round training utilized more exploratory parameters ($T = 1.1$, top-p = 1.0) with 200 sequences per structure to encourage broader search space exploration. For **evaluation**, temperature was reduced to $T = 0.15$ with 128 sequences per structure to ensure reproducible comparisons across benchmarks.

## A.2 ANTIMICROBIAL PEPTIDE DESIGN

**Training Details**  Following (Cao et al., 2025b), the PPO framework applied learning rate $1 \times 10^{-5}$, increased batch size to 256, and reduced training to 10 epochs over 3000 steps. The MIC predictor employed ESM2 embeddings processed through multi-layer perceptron architecture, trained using Focal Loss with focusing parameter $\gamma$ and class weighting $\alpha_i$ to address dataset imbalance. The base ProGen2-xlarge model (6.4B parameters) underwent supervised fine-tuning using LoRA adaptation with rank 32, alpha 16, dropout 0.1, learning rate $1 \times 10^{-4}$, and batch size 16 over 30 epochs.

**Sampling Details**  Sequence generation employed temperature 1.0, top-p 0.95, beam number 4, length penalty 1.2, and repetition penalty 1.2 with maximum sequence length 50. It also excluded invalid amino acids (B, O, U, X, Z) and maintained consistent sequence length constraints. The sampling strategy ensured diverse peptide generation while preserving antimicrobial sequence patterns learned during fine-tuning. During evaluation, 131,072 (4096*32) and 160,000 AMP sequences are generated for pass@k-related metric and latent visualization, respectively.

## A.3 KINASE MUTATION

**Training details**  Following (Wang et al., 2024), we applied learning rate $1e-5$, batch 16. We set the maximum total steps to 10,000 and created 50 parallel environments for training. The entropy-loss coefficient is set to 0. To prevent the policy from being trapped in local optima, we set the discount factor to 0 so that only the reward of the final sequence is used. The PPO clipping ratio is 0.2 and the sampling temperature is 1.0. Reward is the experimentally measured fitness from the dataset; sequences not present in the dataset receive -1, and invalid sequences receive -100.

**Sampling details**  During sampling, we used temperature = 1.0 and top-p = 1.0, and created a single environment to perform 50,000 rounds of sampling. Following the same protocol as in training, mutation was terminated—and the result saved—either when the maximum number of mutation steps was reached or when the sequence's fitness exceeded the initial fitness. After each sample was completed, the environment was re-initialized by randomly selecting a new starting sequence from the test set for the next round of mutation.

## A.4 ANTIBODY MUTATION

**Training details**  During training, we used a fixed random seed as 42 and optimize with Adam (lr=$4e-5$, weight decay=$1e-4$), batch size 32, and global gradient clipping at 0.5 for 30 epochs. Training is conducted on 4 A100 GPUs. In RL fine-tuning, we performed on-policy multi-step rollouts ($T = 4$) and restrict edits to CDR-masked positions. At each step, up to four sites are mutated while disallowing the wild-type residue; sites are selected greedily from position probabilities, and amino acids are chosen as the non-wild-type $\arg\max$ under temperature-scaled logits. Both amino-acid and position temperatures linearly anneal from 1.0 to 0.5 over the first 1,000 steps. Advantages/returns are computed with GAE ($\gamma = 0.99$, $\lambda = 0.95$), returns are standardized, and GRPO rank normalization is applied to advantages. We optimize a PPO objective with clipping 0.2; the log-probability sums amino-acid and position terms with weight 0.5 on the position term:

$$\log \pi = \log \pi_{\text{AA}} + 0.5 \log \pi_{\text{pos}}.$$

The loss weight for KL loss ($\alpha$), value loss ($\beta$), and entropy loss ($\gamma$) are 20.0, 0.4, and 0.01, respectively. KL was computed only at mutated sites; and the final KL term is clipped to $\leq 10.0$. Sequences and masks use `pad_id`$= 1$.

**Sampling details**   We generate each antibody sequence we mutate up to $K$ sites (default $K = 4$); site choice and residue replacement are driven by the model's position propensities and amino-acid logits with `temperature`=1.0, `position_temp`=1.0, and `position_threshold`=0.5. A frozen reward model predicted ddG for each mutant and can optionally score the wild type. Inference was conducted on an single A100 GPU, default `batch_size`=16.

## B   DATASET AND EVALUATION METRICS

### B.1   KINASE MUTATION

**Datasets**   We evaluate the impact of RL training on protein mutations using the PhoQ dataset(Podgornaia & Laub (2015)). This dataset provides 140,517 annotated data points among the 160,000 possible variants that differ at four mutational sites (A284,V285, S288, T289). Fitness is reported as the corresponding phosphatase or kinase activity for each PhoQ variant. For the re-maining unlabeled variants, we follow the convention in Wang et al. (2024) and assign a fitness value of -1. Following the fitness-split protocol of (Ouyang-Zhang et al. (2023)), we fixed the fourth site and partitioned the first three positions into training and test sets in an 8:2 ratio. All sequences were then assigned to four bins according to their fitness values ($=0, \leq 1, \leq 10, >10$). From each bin we randomly sampled 100 sequences in both the training and test splits to form the initial-seed pools.

**Evaluation metrics**   **Positional entropy** is calculated as the Shannon entropy at each of the first three mutable positions of the mutated sequence.

### B.2   ANTIBODY MUTATION

**Datasets**   we utilized AB1101, an open-source dataset comprising 32 antigen-antibody complexes with comprehensive mutational sequence data. This dataset contains 645 single-point mutation entries and 456 multi-point mutation entries. Following the data partitioning strategy established in ProtAttBA, we employed single-point mutations as training data for both the policy model and reward model, while reserving multi-point mutations for testing purposes. For reinforcement learning training, we performed additional stratification of the multi-point mutation data based on complex PDB identifiers. Through random selection, we designated 1MLC and 1VFB as the test PDB structures to ensure robust evaluation of our approach.

**Evaluation metrics**   **Positional Entropy** is calculated based on the Shannon entropy at each individual position within the CDR (Complementarity-Determining Region) of the mutated antibody sequences. **Perplexity** is computed as the exponential of the average negative log-likelihood of the generated protein sequences under a specific model.

### B.3   AMP DESIGN

**Datasets**   Following the AMPHION framework, we trained a reward model utilizing currently available open-source MIC values sourced from three established databases: DBAASP, DRAMP, and APD3. We selected sequences with lengths ranging from 6 to 50 amino acids and classified them as active peptides based on MIC values below 32 $\mu$g/mL. This process yielded a total of 7,888 samples encompassing both positive and negative instances. To ensure proper data partitioning and minimize sequence similarity bias, we employed MMseqs2 for clustering and dataset splitting, resulting in a training, validation, and test distribution of 6153, 789, and 946 samples, respectively.

**Evaluation metrics**   **Diversity** is calculated as the average sequence similarity between generated sequences, measuring the model's ability to produce varied outputs and avoid mode collapse. **Novelty** is computed based on the degree of novelty between generated sequences and natural sequences, specifically defined as the average of (1 - maximum sequence similarity) across all generated sequences, where the maximum similarity represents the highest identity score between each generated sequence and any reference natural sequence. **Perplexity** is computed as the exponential of the average negative log-likelihood of the generated protein sequences under a specific model.

## B.4 PROTEIN INVERSE FOLDING

**Datasets** The dataset construction leverages CATH4.2, following to the official train-test split scheme established by the CATH database. Both the base model training and DPO dataset construction are exclusively conducted on the training partition, which comprises 18,024 protein structures. Model performance evaluation is carried out on the CATH4.2 test set containing 1,120 structures, supplemented by two additional evaluation benchmarks: TS50 and TS500, which contain 50 and 470 protein structures respectively. This evaluation framework ensures comprehensive assessment across diverse structural complexity and provides robust validation of the model's generalization capabilities.

**Evaluation metrics** **Diversity** is calculated as the average sequence similarity between generated sequences, measuring the model's ability to produce varied outputs and avoid mode collapse. **Novelty** is computed based on the degree of novelty between generated sequences and natural sequences, specifically defined as the average of (1 - maximum sequence similarity) across all generated sequences, where the maximum similarity represents the highest identity score between each generated sequence and any reference natural sequence. **Perplexity** is computed as the exponential of the average negative log-likelihood of the generated protein sequences under a specific model. **Recovery Rate** is computed as the percentage of amino acid positions that are correctly predicted compared to the native sequence. **TM-Score** evaluates the structural similarity between predicted and native structures by measuring the geometric alignment quality across all residue positions.

# C    METHOD DETAILS

## C.1    ANTIBODY MUTATION NETWORK

---

**Algorithm 1** ProtAttBA-improved: Antibody-antigen Binding Affinity Prediction and Featurization

---

**Require:** Wild-type antibody sequence $S_{ab}^{wt}$, mutant antibody sequence $S_{ab}^{mt}$
**Require:** Wild-type antigen sequence $S_{ag}^{wt}$, mutant antigen sequence $S_{ag}^{mt}$
**Require:** Attention masks $M_{ab}^{wt}, M_{ab}^{mt}, M_{ag}^{wt}, M_{ag}^{mt}$
**Ensure:** Binding affinity prediction $\hat{y}$ and auxiliary outputs
1: **// Step 1: Protein Language Model Encoding**
2: $E_{ab}^{wt} \leftarrow \text{ESM}(S_{ab}^{wt}, M_{ab}^{wt}) \in \mathbb{R}^{B \times L_{ab} \times H}$ {Wild-type antibody embeddings}
3: $E_{ab}^{mt} \leftarrow \text{ESM}(S_{ab}^{mt}, M_{ab}^{mt}) \in \mathbb{R}^{B \times L_{ab} \times H}$ {Mutant antibody embeddings}
4: $E_{ag}^{wt} \leftarrow \text{ESM}(S_{ag}^{wt}, M_{ag}^{wt}) \in \mathbb{R}^{B \times L_{ag} \times H}$ {Wild-type antigen embeddings}
5: $E_{ag}^{mt} \leftarrow \text{ESM}(S_{ag}^{mt}, M_{ag}^{mt}) \in \mathbb{R}^{B \times L_{ag} \times H}$ {Mutant antigen embeddings}
6: **// Step 2: Attention-based Feature Enhancement**
7: **for** $x \in \{E_{ab}^{wt}, E_{ab}^{mt}, E_{ag}^{wt}, E_{ag}^{mt}\}$ **do**
8:    $x' \leftarrow \text{AttnTransform}(x) = \text{softmax}(\text{Conv1D}(\text{LayerNorm}(x))) \odot x$ {Enhanced features}
9: **end for**
10: **// Step 3: Cross-Modal Attention Mechanism**
11: $\tilde{E}_{ag}^{wt} \leftarrow \text{MultiHeadAttn}(E_{ag}^{wt}, E_{ab}^{wt}, E_{ab}^{wt}, M_{ab}^{wt})$ {Antigen attends to antibody}
12: $\tilde{E}_{ag}^{mt} \leftarrow \text{MultiHeadAttn}(E_{ag}^{mt}, E_{ab}^{mt}, E_{ab}^{mt}, M_{ab}^{mt})$ {Mutant antigen-antibody attention}
13: $\tilde{E}_{ab}^{wt} \leftarrow \text{MultiHeadAttn}(E_{ab}^{wt}, E_{ag}^{wt}, E_{ag}^{wt}, M_{ag}^{wt})$ {Antibody attends to antigen}
14: $\tilde{E}_{ab}^{mt} \leftarrow \text{MultiHeadAttn}(E_{ab}^{mt}, E_{ag}^{mt}, E_{ag}^{mt}, M_{ag}^{mt})$ {Mutant antibody-antigen attention}
15: **// Step 4: Auxiliary Classification Heads**
16: $L_{ab}^{wt} \leftarrow \text{Linear}(\tilde{E}_{ab}^{wt}) \in \mathbb{R}^{B \times L_{ab} \times 33}$ {Wild-type antibody logits}
17: $L_{ab}^{mt} \leftarrow \text{Linear}(\tilde{E}_{ab}^{mt}) \in \mathbb{R}^{B \times L_{ab} \times 33}$ {Mutant antibody logits}
18: $L_{ag}^{wt} \leftarrow \text{Linear}(\tilde{E}_{ag}^{wt}) \in \mathbb{R}^{B \times L_{ag} \times 33}$ {Wild-type antigen logits}
19: $L_{ag}^{mt} \leftarrow \text{Linear}(\tilde{E}_{ag}^{mt}) \in \mathbb{R}^{B \times L_{ag} \times 33}$ {Mutant antigen logits}
20: **// Step 5: Attention-weighted Global Pooling**
21: $h_{ab}^{wt} \leftarrow \text{AttnMean}(\tilde{E}_{ab}^{wt}, M_{ab}^{wt}) \in \mathbb{R}^{B \times H}$ {Pooled antibody representation}
22: $h_{ab}^{mt} \leftarrow \text{AttnMean}(\tilde{E}_{ab}^{mt}, M_{ab}^{mt}) \in \mathbb{R}^{B \times H}$ {Pooled mutant antibody}
23: $h_{ag}^{wt} \leftarrow \text{AttnMean}(\tilde{E}_{ag}^{wt}, M_{ag}^{wt}) \in \mathbb{R}^{B \times H}$ {Pooled antigen representation}
24: $h_{ag}^{mt} \leftarrow \text{AttnMean}(\tilde{E}_{ag}^{mt}, M_{ag}^{mt}) \in \mathbb{R}^{B \times H}$ {Pooled mutant antigen}
25: **// Step 6: Complex Formation and Prediction**
26: $h^{wt} \leftarrow h_{ab}^{wt} + h_{ag}^{wt}$ {Wild-type complex representation}
27: $h^{mt} \leftarrow h_{ab}^{mt} + h_{ag}^{mt}$ {Mutant complex representation}
28: $h_{concat} \leftarrow \text{Concat}(h^{wt}, h^{mt}) \in \mathbb{R}^{B \times 2H}$ {Concatenated complexes}
29: $h_{norm} \leftarrow \text{BatchNorm}(h_{concat})$ {Normalized features}
30: **// Step 7: Multi-layer Prediction Head**
31: $h_1 \leftarrow \text{Tanh}(\text{Linear}(h_{norm})) \in \mathbb{R}^{B \times H/2}$ {First hidden layer}
32: $h_1 \leftarrow \text{Dropout}(h_1, p = 0.1)$ {Apply dropout}
33: $h_2 \leftarrow \text{ReLU}(\text{Linear}(h_1)) \in \mathbb{R}^{B \times H/2}$ {Second hidden layer}
34: $\hat{y} \leftarrow \text{Linear}(h_2) \in \mathbb{R}^{B}$ {Binding affinity prediction}
35: **return** $\hat{y}, L_{ab}^{wt}, L_{ag}^{wt}, L_{ab}^{mt}, L_{ag}^{mt}$

---

## C.2    ANTIBODY MUTATION STRATEGY

---

**Algorithm 2** Policy-Guided Antibody Mutation

---

**Require:** Policy model position probabilities $P_{pos} \in \mathbb{R}^L$
**Require:** Mutation model logits $L_{mut} \in \mathbb{R}^{L \times V}$
**Require:** Wild-type sequence $S_{wt}$, CDR mask $M_{cdr}$
**Require:** Temperature $\tau$, stochastic flag $s$
 1: $P_{masked} \leftarrow P_{pos} \odot M_{cdr}$ {Mask to CDR}
 2: **Position Selection:**
 3: **if** $s = $ **True then**
 4:    Select positions where $P_{masked} > \theta$ via multinomial sampling {**Stochastic inference**}
 5: **else**
 6:    Select top-$k$ positions by $P_{masked}$ values {**Deterministic training**}
 7: **end if**
 8: **Amino Acid Mutation:**
 9: **for** each selected position $i$ **do**
10:    $P_{aa} \leftarrow \text{softmax}(L_{mut}[i]/\tau)$ {Apply temperature}
11:    $P_{aa}[S_{wt}[i]] \leftarrow 0$ {Mask wild-type residue}
12:    $P_{aa} \leftarrow P_{aa}/\sum P_{aa}$ {Re-normalize}
13:    **if** $s = True$ **then**
14:       $S_{mut}[i] \leftarrow \text{sample}(P_{aa})$ {Stochastic sampling}
15:    **else**
16:       $S_{mut}[i] \leftarrow \arg\max(P_{aa})$ {Greedy selection}
17:    **end if**
18: **end for**
19: **return** Mutated sequence $S_{mut}$

---

## C.3 KINASE MUTATION STRATEGY

---

**Algorithm 3** Kinase Mutation

---

**Require:** Initial sequence $S_0 \in \mathbb{Z}^L$, Pretrained ESM Encoder $f_{ESM}$, Action Network $f_{Action}$,
   Value Network $f_{Value}$, Masked Language Model $f_{MLM}$, Tokenizer $T$, Mutation positions $P = \{p_1, p_2, p_3\}$, Amino acid vocabulary $\mathbb{A}$.
**Ensure:** Final mutated sequence $S_N$.
 1: **// Initialization**
 2: $S_{obs} \leftarrow S_0$ {Initialize observation with the starting sequence}
 3: $E_{ref} \leftarrow f_{ESM}(S_{protein})$ {Get reference embedding for normalization}
 4: **for** $t = 0$ to $N - 1$ **do**
 5:    **// Step 1: Sequence Feature Extraction**
 6:    $E_t \leftarrow f_{ESM}(S_{obs}) \in \mathbb{R}^{B \times L \times H}$ {Encode current sequence}
 7:    $E'_t \leftarrow E_t/(E_{ref} + \epsilon)$ {Normalize embeddings}
 8:    $E_{flat} \leftarrow \text{Flatten}(E'_t) \in \mathbb{R}^{B \times (L \cdot H)}$ {Flatten for policy networks}
 9:    **// Step 2: Select Mutation Position (Actor)**
10:    $\mathbf{l}_{pos} \leftarrow f_{Action}(E_{flat}) \in \mathbb{R}^{B \times |P|}$ {Get logits for positions}
11:    $\pi_{pos} \leftarrow \text{Categorical}(\text{logits} = \mathbf{l}_{pos})$ {Create position distribution}
12:    $idx_p \leftarrow \pi_{pos}.\text{sample}()$ {Sample position index, e.g., 0, 1, or 2}
13:    $p_t \leftarrow P[idx_p]$ {Map index to actual sequence position, e.g., 96, 97, 100}
14:    **// Step 3: Predict Candidate Amino Acid (Masked LM)**
15:    $S_{mask} \leftarrow S_{obs}; S_{mask}[p_t] \leftarrow \text{MASK\_TOKEN}$ {Mask selected position}
16:    $\mathbf{l}_{aa} \leftarrow f_{MLM}(S_{mask})[p_t] \in \mathbb{R}^{|\mathbb{A}|}$ {Get logits for amino acids at position $p_t$}
17:    $\pi_{aa} \leftarrow \text{Softmax}(\mathbf{l}_{aa}/\tau)$ {Create amino acid distribution with temperature $\tau$}
18:    $a_t \leftarrow \pi_{aa}.\text{sample}()$ {Sample a new amino acid token}
19:    **// Step 4: Update Sequence and Get Reward**
20:    $S_{new} \leftarrow S_{obs}; S_{new}[p_t] \leftarrow a_t$ {Apply mutation}
21:    $R_t, \text{done} \leftarrow \text{PhoQEnv.step}(S_{new})$ {Get reward from environment}
22:    $S_{obs} \leftarrow S_{new}$ {Update the state for the next iteration}
23: **end for**
24: **return** $S_{obs}$

---

## C.4 Reinforcement Learning Loss for AMP Design

We employ three widely used RL algorithms to fine-tune PLMs $p_\theta(\mathbf{s})$ parameterized by $\theta$, each targeting different aspects of biological knowledge acquisition.

**Direct Preference Optimization (DPO)** DPO learns from preference pairs without explicit reward modeling. Given preference dataset $\mathcal{D} = \{(\mathbf{s}_i^+, \mathbf{s}_i^-)\}$ where $\mathbf{s}_i^+ \succ \mathbf{s}_i^-$, the DPO loss is:

$$\mathcal{L}_{DPO}(\theta) = -\mathbb{E}_{(\mathbf{s}^+, \mathbf{s}^-) \sim \mathcal{D}} \left[ \log \sigma \left( \beta \log \frac{p_\theta(\mathbf{s}^+)}{p_{ref}(\mathbf{s}^+)} - \beta \log \frac{p_\theta(\mathbf{s}^-)}{p_{ref}(\mathbf{s}^-)} \right) \right],$$

where $p_{ref}$ is the reference model, $\beta > 0$ controls KL divergence, and $\sigma$ is the sigmoid function.

**Proximal Policy Optimization (PPO)** PPO optimizes the policy using clipped importance sampling. For sequence $\mathbf{s}$ with reward

$$\mathcal{L}_{\text{PPO}}(\theta) = \mathbb{E}_{\mathbf{s} \sim p_{\theta_{\text{old}}}} \left[ \min \left( \rho_\theta(\mathbf{s}) \hat{A}(\mathbf{s}), \text{clip}(\rho_\theta(\mathbf{s}), 1 - \epsilon, 1 + \epsilon) \hat{A}(\mathbf{s}) \right) \right] + \beta \cdot \mathbb{E}_{\mathbf{s} \sim p_\theta} \left[ D_{\text{KL}}(p_\theta(\mathbf{s}) || p_{\theta_{\text{old}}}(\mathbf{s})) \right],$$

where $\rho_\theta(\mathbf{s}) = \frac{p_\theta(\mathbf{s})}{p_{\theta_{\text{old}}}(\mathbf{s})}$ represents the **importance ratio**, which measures the change in probability of a sequence between the current and the old policy. The term $\hat{A}(\mathbf{s})$ is the **advantage estimate**, which quantifies how much better or worse the current action is compared to the baseline. The parameter $\epsilon$ is the **clipping parameter**, which ensures that the policy update does not change too drastically. To prevent large policy updates, the loss includes the **Kullback-Leibler (KL) divergence**, denoted as $D_{\text{KL}}(p_\theta(\mathbf{s}) || p_{\theta_{\text{old}}}(\mathbf{s}))$, which measures deviation from the old policy to the current policy. Finally, $\beta$ denotes a **hyperparameter** that controls the strength of the KL regularization.

**Group Relative Policy Optimization (GRPO)** GRPO computes relative advantages within sequence groups, making it suitable for comparative protein design. For group $\mathcal{G} = \{\mathbf{s}_1, \ldots, \mathbf{s}_m\}$, the relative advantage is:

$$\hat{A}^{rel}(\mathbf{s}_j) = R(\mathbf{s}_j) - \frac{1}{m} \sum_{i=1}^{m} R(\mathbf{s}_i)$$

The GRPO loss extends PPO with group-wise normalization:

$$\mathcal{L}_{\text{GRPO}}(\theta) = \mathbb{E}_{\mathcal{G}} \left[ \frac{1}{|\mathcal{G}|} \sum_{\mathbf{s} \in \mathcal{G}} \min \left( \rho_\theta(\mathbf{s}) \hat{A}^{rel}(\mathbf{s}), \text{clip}(\rho_\theta(\mathbf{s}), 1 - \epsilon, 1 + \epsilon) \hat{A}^{rel}(\mathbf{s}) \right) \right]$$

$$+ \gamma \cdot \mathbb{E}_{\mathbf{s} \sim \pi_\theta} \left[ D_{\text{KL}}(p_\theta(\mathbf{s}) || p_{\theta_{\text{old}}}(\mathbf{s})) \right],$$

where $\hat{A}^{rel}(\mathbf{s}_j)$ represents the **relative advantage** of sequence $\mathbf{s}_j$ within the group $\mathcal{G}$. The importance ratio, denoted as $\rho_\theta(\mathbf{s}) = \frac{p_\theta(\mathbf{s})}{p_{\theta_{\text{old}}}(\mathbf{s})}$, measures the change in probability of a sequence between the current and old policies. The clipping parameter $\epsilon$ ensures that the policy update remains within a bounded range, preventing large, destabilizing changes. The **KL divergence** is regularized to avoid large policy updates. Finally, $\gamma$ denotes the hyperparameter that controls the strength of the KL regularization.

## D SUPPLEMENTARY EXPERIMENTAL RESULTS OF ANTIBODY MUTATION

### D.1 TEST RESULTS ON THE RE-IMPLEMENTED PROTATTBA

| Method | RMSE | Pearson cor. | Spearman cor. |
|---|---|---|---|
| ProtAttBA (Original) | 2.10 | 0.55 | 0.45 |
| ProtAttBA (Ours) | **1.50** | **0.58** | **0.47** |

### D.2 DISTRIBUTION SHIFT OF PREDICTED DDG

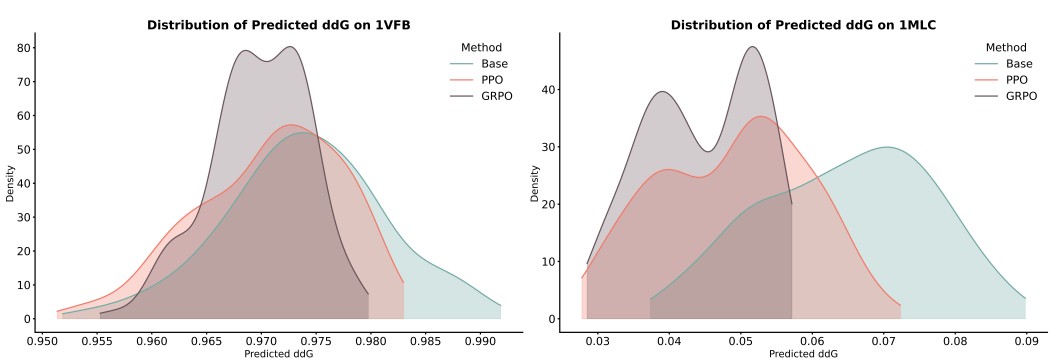

Figure 6: Predicted ddG distribution on test set PDBs.

# E   SUPPLEMENTARY EXPERIMENTAL RESULTS OF INVERSE FOLDING

## E.1   ABLATION ON SUPPORT METRIC OF K

Table 2: Results for *Support* metric for four biological systems.

|  | **Preservation** | **Expansion** | **Shrinkage** | **Out-Of-Support** | **ESR ↑** |
|---|---|---|---|---|---|
| k=128 | 891 | 9 | 21 | 199 | 2.33 |
| k=32 | 886 | 12 | 23 | 199 | 1.92 |
| k=8 | 865 | 23 | 27 | 205 | 1.17 |
| k=2 | 831 | 31 | 33 | 225 | 1.06 |

## E.2   LATENT VISUALIZATION OF SUPPORT SUBSETS

# F   SUPPLEMENTARY EXPERIMENTAL RESULTS OF AMP DESIGN

## F.1   BINARY CLASSIFICATION PERFORMANCE OF REWARD MODEL

Table 3: Results for ApexMIC on binary classification.

|  | Accuracy | Precision | Sensitivity | Specificity | F1-score | AUC-ROC |
|---|---|---|---|---|---|---|
| ApexMIC | 0.96 | 0.62 | 0.82 | 0.98 | 0.70 | 0.90 |

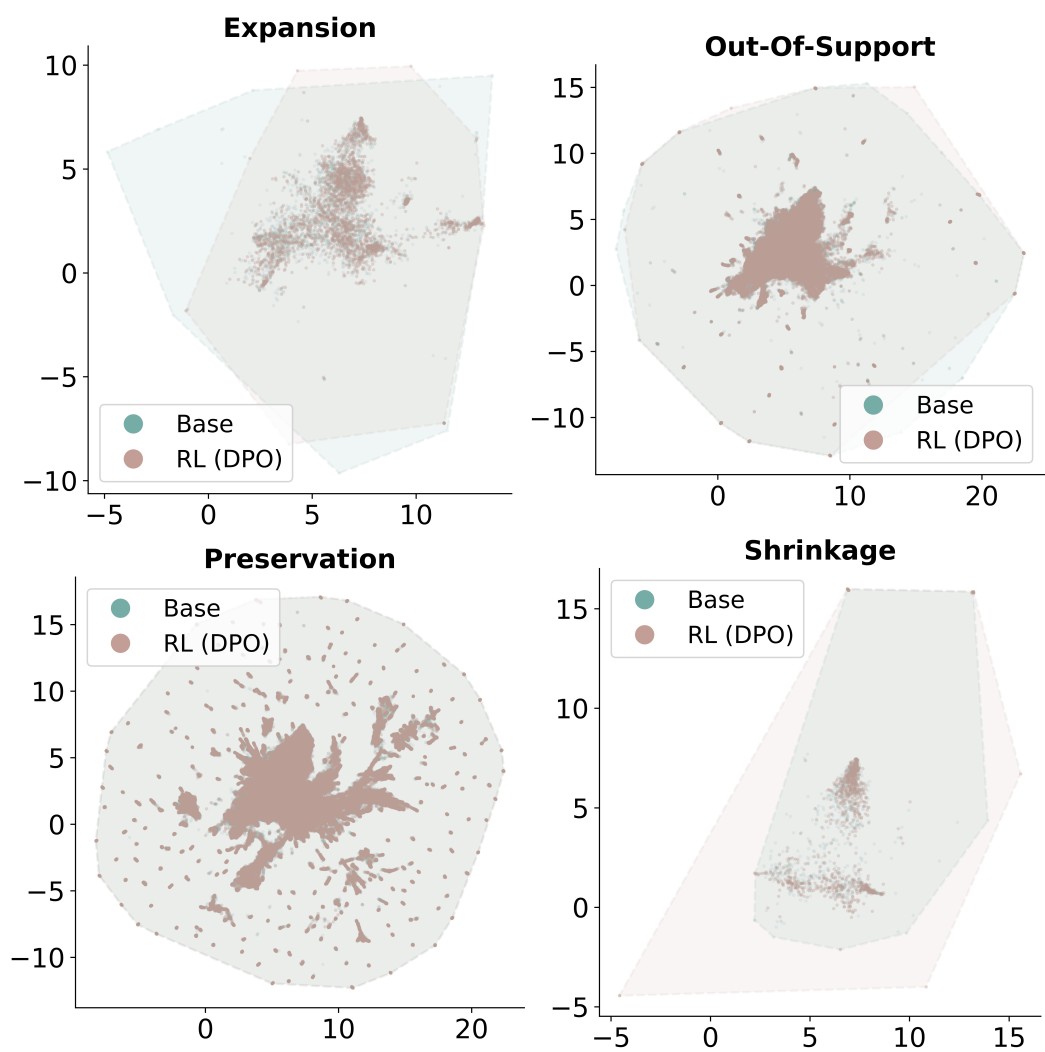

Figure 7: Latent visualization on inverse folding tasks for different support subsets.

## G  SUMMARY OF REINFORCEMENT LEARNING-GUIDED PROTEIN DESIGN METHODS

Table 4: Taxonomy of RL approaches in protein design.

| Task | Method | RL Algorithm | Reward Function | Designed Biological Entity |
|------|--------|--------------|-----------------|----------------------------|
| **Structural Design** | Issac et al. (Lutz et al., 2023) | MCTS | Composite structural score: architecture/topology fit, sterics, geometry constraints | Symmetric multi-subunit protein assemblies |
| | | | | *Continued on next page* |

| Task | Method / Paper | RL Algorithm | Reward Function | Designed Entity |
|------|----------------|--------------|-----------------|-----------------|
| | GAPN (Gao et al., 2024) | PPO | Direct docking reward (pose/energy), adversarial reward to improve global assembly rules | Multimer protein complex assembly / docking paths |
| **Sequence Optimization** | EvoPlay (Wang et al., 2023) | MCTS (AlphaZero) | Task-specific predicted fitness/activity (surrogate property predictors) | Enzymes / general proteins |
| | RLXF (Blalock et al., 2025) | PPO | Experimentally measured function (e.g., fluorescence/fitness) | Diverse protein families (e.g., CreiLOV variants) |
| | μProtein (Sun et al., 2025) | PPO | Predicted/experimental fitness from landscape model | Enzymes (e.g., $\beta$-lactamase) |
| | RelaVDEP (Mi et al., 2025) | MCTS | Fine-tuned SPIRED-Fitness; structure/foldability filters (ESMFold/AF2 pLDDT, SPIRED-Stab $\Delta\Delta G$/$\Delta T_m$); diversity–fitness metric | GFP (fluorescence), NUDT15/VKOR1 (cellular abundance), AmeR (fold repression), PETase (enzymatic activity) |
| | ApexAmphion (Cao et al., 2025b) | PPO | Predicted MIC, physicochemical properties | Board-spectrum antimicrobial peptides |
| **Inverse Folding** | ProteinZero (Wang et al., 2025b) | PPO, GRPO | ESM-Fold structural fidelity, $\Delta\Delta G$ stability proxy, diversity | General protein sequences |
| | Xu et al. (Xu et al., 2025) | Multi-round DPO | TM-score | General protein sequences |
| | EnerBridge-DPO (Rong et al., 2025) | DPO | Energy score | Protein complex sequences |
| | ResiDPO (Xue et al., 2025) | DPO | pLDDT score, designability | General protein sequences |
| | Park et al. (Park et al., 2024) | DPO | TM-score, diversity metric | Peptide / short-protein sequences |
| | RL-DIF (Ektefaie et al., 2024) | DDPO | Foldability, TM-score | General protein sequences |
| | ProtInvTree (Liu et al., 2025b) | MCTS | TM-score, scTM-score from ESMFold | General protein sequences |

*Continued on next page*

| Task | Method / Paper | RL Algorithm | Reward Function | Designed Entity |
|------|----------------|--------------|-----------------|-----------------|
| | DRAKES (Wang et al.) | Reward Fine-tuning | Sequence stability | General protein sequences |
| | GLID$^2$E (Cao et al., 2025a) | PPO | Sequence stability | General protein sequences |
| **Antibody Engineering** | AB-Gen (Xu et al., 2023b) | REINVENT | Developability, specificity | Antibody CDRH3 libraries (HER2, etc.) |
| | BetterBodies (Vogt et al., 2024) | Q-learning | Absolute free-energy, affinity | Antibody CDRH3 binders (SARS-CoV-2 RBD, etc.) |
| | Structured Q-learning (Cowen-Rivers et al., 2022) | Q-learning | Docking affinity | Antibody CDRH3 binders (IGG4, etc.) |
| **Peptide Binder Design** | TCRPPO (Chen et al., 2023) | PPO | Valid-TCR likelihood, peptide-recognition probability | T-cell receptor (TCR) sequences ($\beta$-chain CDR3, etc.) |
| | HighPlay (Lin et al., 2025) | MCTS | Structure-/pose-guided scores, binding/energy proxies by HighFold | Cyclic peptide binders |
| | CYC_BUILDER (Wang et al., 2025a) | MCTS | Docking/binding-energy, pose-quality scores | Cyclic peptide binders |

## H MORE RELATED WORK

### H.1 PROTEIN LANGUAGE MODELS AND BEYOND

The landscape of protein language models compose distinct architectures: BERT-based encoder models like ESM-2 (Lin et al., 2023) excel at understanding tasks through bidirectional context, autoregressive models such as ProGen2 (Nijkamp et al., 2023) and ProtGPT2 (Ferruz et al., 2022) focus on generation, while recent approaches like ESM-3 (Hayes et al., 2024), SaProt (Su et al., 2023), and xTrimoPGLM (Chen et al., 2024) integrate multimodal information. A critical limitation of these foundational models is their general focus, which delivers diminishing returns for specialized protein tasks despite requiring substantial computational resources. This has driven emergence of protein-specific architectures including antibody models like IgLM (Shuai et al., 2023) and AbLang (Olsen et al., 2022), enzyme systems like ZymCTRL (Munsamy et al., 2024), and domain-targeted approaches for inverse folding (Qiu et al., 2024), RL design (Cao et al., 2025b) and membrane proteins (Zhang et al., 2024d). These specialized models outperform general approaches through domain-specific training, but face fundamental limitations on training data.

### H.2 RL FOR NATURAL LANGUAGE PROCESSING

Math reasoning is a key RL success in NLP, showing emergent self-verification and adaptive scaling via GRPO and RL with Verifiable Rewards (DeepSeek-AI, 2025; Zhang et al., 2024b; Cobbe et al., 2021). Multimodal tasks use RL for cross-modal reasoning; MAYE and RLHF-V help vision-language models solve math and reduce hallucinations (Wu et al., 2024; Liu et al., 2024). RL is also used for compiler feedback (Zhang et al., 2024a), conversational optimization (Zhang et al., 2024c),

and ranking (Paulus et al., 2017; Ranzato et al., 2015). The effectiveness of RLVR for reasoning is contested: some view it as smart sampling toward high-reward outputs (Gandhi et al., 2025; Shah et al., 2025); several studies attribute reasoning to pretraining (Yue et al., 2025; Wu et al., 2025; Wu & Choi) and argue RLVR echoes pretrained patterns (Zhao et al., 2025). Others report gains from structured RLVR (Liu et al., 2025c) and from unlikeliness rewards to reduce rank bias (He et al., 2025). Wen et al. (2025) propose CoT-`passk`, showing RLVR benefits under more robust evaluation.

# I  USAGE OF LANGUAGE MODELS

We use large language model (LLM) to aid in the preparation of this manuscript. Its use was limited to editorial tasks, including proofreading for typographical errors, correcting grammar, and improving the clarity and readability of the text.

