# OpenReview forum: "From Supervision to Exploration: What Does Protein Language Model Learn During Reinforcement Learning?"
_ICLR.cc/2026/Conference — ICLR 2026 Conference Desk Rejected Submission_

### Official Review · Reviewer_k7Wy · 2025-10-30

**Soundness:** 3
**Presentation:** 4
**Contribution:** 3
**Rating:** 4
**Confidence:** 5

**Summary:**

This paper presents a comprehensive study investigating what reinforcement learning (RL) teaches Protein Language Models (PLMs) during the fine-tuning process for protein design. The central question is whether RL enables PLMs to discover new, emergent capabilities beyond their pre-training data, or simply amplifies existing patterns. Through an analysis of two leading PLM architectures, three RL algorithms, and four prominent experimental systems, this paper concluded that RL enables more efficient sampling of high-reward regions. However, RL’s ability to learn new patterns and optimize high-reward distributions comes with trade-offs. These effects are influenced by factors such as the initialization capacity of the policy model (base model), reward accuracy, and task complexity.

**Strengths:**

1. The paper is trying to answer a critical and timely question in the field: " Do new emergent capabilities arise during the RL fine-tuning process of PLMs?"
2. This work systematically evaluates RL-enhanced PLMs across multiple biological systems and reports consistent improvements in sampling efficiency and optimization performance.
3. This paper provides a clear guidance. The performance is influenced by three factors such as the initialization capacity of the policy model (base model), reward accuracy, and task complexity.

**Weaknesses:**

1. The exploration of policy capacity is relatively shallow. Policy capacity is described as a combination of model size and representational power, but it's not systematically quantified or varied. The study uses different base PLMs for different tasks, making it difficult to isolate the effect of capacity alone. A more rigorous test would involve using the same RL algorithm and reward on the same task while systematically scaling the size of the policy model (e.g., from 100M to 7B parameters) to directly show how gains scale with capacity.
2. The paper answers what RL does (efficient sampling) but provides less insight into how it does it or what specific knowledge is encoded. For example, while latent space visualizations (UMAP) show distributional shifts, they don't explain which sequence-structure-function rules the model has internalized. Does it learn physic-chemical principles, conserved motifs, or folding constraints? The study identifies the effect (prioritizing high-TM-score regions) but not the mechanism. A deeper analysis linking the RL-guided changes to specific, interpretable biological features (e.g., changes in hydrophobicity, charge distribution, or known functional sites) would be more compelling.
3. It's unclear if the framework holds for more complex, multi-objective tasks (e.g., design a stable, potent, and non-toxic AMP) where reward shaping becomes even more critical.

**Questions:**

1. The paper states that the policy model's initial capacity is a key factor. To what extent are your results simply a reflection of the knowledge already present in the pre-trained base models? Is RL primarily "unlocking" or "focusing" the base model's latent knowledge, rather than teaching it anything fundamentally new about protein biophysics?
2. The paper concludes RL is an "efficient sampler" and does not create "new emergent capabilities." Could this be a matter of definition? If RL successfully guides the model to a functionally novel region of sequence space that the base model could technically reach but whose functional value it did not appreciate, isn't that a form of learning a new "sequence-function relationship"?
3. It seems "Task Difficulty" is often defined by the accuracy of the reward signal and the complexity a policy must capture. Could this be reformulated as a two-factor model: "Signal Quality" (reward accuracy) and "Model Capability" (policy capacity to exploit the signal given the task's complexity)?
4. In practice, creating a highly accurate reward model (e.g., with a Spearman >0.9) is often the most expensive part of the process. How should a practitioner decide on the sufficient level of reward accuracy before investing in larger policy models, given diminishing returns?

---

> ### Author Response · Authors · 2025-11-28
>
> We thank the reviewer for the thoughtful and encouraging assessment. We address the specific weaknesses and questions below.
>
> **Policy Capacity and Scaling**
> > Regarding the suggestion for a controlled scaling study (e.g., 100M to 7B parameters), we agree this would be the definitive test for isolating capacity effects. While computational constraints prevented a full sweep in this work, we operationalized "capacity" via the observable behavior of the base model: using base Pass@k curves, perplexity, and support size as proxies for how much task structure the pre-trained PLM already captures. Our results confirm that higher base capacity (as seen in Inverse Folding) correlates with net expansion (ESR>1), whereas weaker base capacity (as in PhoQ) leads to mode collapse. We will clarify this proxy approach in the text and acknowledge systematic scaling as a priority for future work.
>
> **Mechanistic Interpretability**
> > Regarding the need to link RL-induced changes to specific biological rules (e.g., hydrophobicity, motifs), we agree this is a crucial next step. Our current work focuses on the behavioral/systems-level diagnostic: *what* changes (sampling efficiency, ESR) and *when* (function of reward/capacity). Bridging this gap to mechanistic sequence-function rules requires extensive in silico and wet-lab validation beyond the scope of this conference paper. We will make this distinction explicit: our contribution is diagnosing the reliability and regions of sampling, while decoding the internal biophysical rules remains future work.
>
> **Multi-Objective Tasks**
> > Regarding the framework's applicability to multi-objective settings, we clarify that our protocol (Pass@k, ESR) is agnostic to the objective count, provided a scalar reward (e.g., weighted sum or Pareto scalarization) and a success threshold exist. Extending our empirical study to explore how different scalarization strategies affect RL behavior is an important direction we will highlight in the conclusion.
>
> **"Signal Quality" + "Model Capability" Reframing**
> > We strongly agree with the reviewer's insight that "Task Difficulty" is better conceptualized as the interplay between **Signal Quality** (reward accuracy/noise) and **Model Capability** (policy capacity relative to landscape complexity). This two-factor abstraction is cleaner and more actionable. We will adopt this terminology in the revision to describe our findings, reframing "difficulty" as an emergent property of this interaction rather than an independent variable.
>
> **"Emergent Capabilities" vs. Reweighting**
> > Regarding whether RL "unlocks" or "focuses" knowledge, our results suggest it primarily reweights latent capabilities. However, we agree with the reviewer that "reliable guidance into functionally novel regions" (as seen in our Inverse Folding results) effectively constitutes learning new sequence-function relationships in a behavioral sense. We will refine our language to reflect this nuance: RL acts as an efficient sampler that, in favorable regimes (high signal/high capacity), can indeed uncover functionally novel high-reward regions that were only theoretically accessible before.
>
> **Practical Reward Accuracy**
> > Regarding the practical decision of when to scale policy capacity, our results suggest a "probe first" recipe: first maximize signal quality, then probe with a moderate-capacity policy. If ESR < 1 and diversity collapses, scaling the policy is unlikely to overcome the fundamental signal limitation. We will explicitly state this practical takeaway: very high reward accuracy is ideal but not strictly necessary for gains, but our diagnostics (ESR) can signal when a reward model is too noisy to justify larger models.
>
> Best regards,
>
> Authors

---

### Official Review · Reviewer_SMxQ · 2025-10-31

**Soundness:** 2
**Presentation:** 1
**Contribution:** 2
**Rating:** 2
**Confidence:** 4

**Summary:**

The authors evaluate the application of three RL algorithms, PPO, DPO, and GRPO, for four protein design tasks: inverse folding, antimicrobial peptide design, kinase mutation, and antibody optimization. Their results show that applying RL can lead to improved performance in these tasks, but also impact on other metrics such as diversity and a “shrinkage” metric.

**Strengths:**

1. The authors apply RL to refine pLMs for four different protein design tasks.
2. Results show the improved performance of RL methods compared to the base pLM model.
3. The research question is interesting and timely.

**Weaknesses:**

1. The authors mention that their framework offers a principled way to measure RL-PLM systems, but their evaluation is limited to a descriptive analysis, missing any theoretical analysis regarding these systems.
2. For the 4 tasks in Section 2, it is unclear which methods were implemented by the authors and which are the changes from the main references.
3. The quality of the Figures makes it hard to evaluate Section 3.

**Questions:**

My initial recommendation is rejection based on the weaknesses mentioned above. My detailed comments are as follows.

Comments:

1. (lines 103-106) In the Introduction section, three key factors are mentioned: task difficulty, reward model accuracy, and policy model capacity. However, limited discussions are made in Section 3 to justify how these factors are chosen.
2. (lines 111-113) The authors mention a “principled way to measure the current RL-PLM systems” but the results are analyzed in a descriptive manner. Which methodology differences are made in the manuscript evaluation that support this sentence and that can be used for future RL-PLM systems?
3. Section 2: It is unclear which parts are taken from references, which methods are re-implemented, and the different characteristics of rewards and surrogates between different tasks, so that it is hard to draw conclusions about the performance of different methods in Section 3.
4. RL methods with different characteristics are evaluated: PPO, DDPO, GRPO. Their different characteristics influence the rewards that can be used, their training methodology, etc. More discussion regarding their differences is needed for a principled evaluation approach.
5. What does the axis mean in Fig. 3?
6. The quality of figures should be improved. The readability of Figures 4 and 5 is limited, which makes it hard to evaluate the results described in Section 3.
7. How can the results in Table 1 be interpreted? Are they averages for different RL methods?
8. For the inverse folding section 3.2, is ESMFold used both as a reward function and as the structure predictor used for evaluation?

Minor Comments (that did not impact the score):

1. Missing the definition of the acronym RL in line 81.
2. The writing quality of the manuscript is weak, with arguable terms like “transcend” and “gains additional power”.
3. Code is not available.
4. The paragraphs in lines 426-431 and lines 432-439 seem redundant.

---

> ### Author Response · Authors · 2025-11-28
>
> We thank the reviewer for the careful and constructive feedback. We address the specific concerns below.
>
> **"Principled" Framework vs. Descriptive Analysis**
> > Regarding the comment that the analysis is descriptive rather than theoretical, we clarify that our intent was not to claim new mathematical RL theory, but to introduce a **unified and reusable evaluation protocol** for RL-PLM systems. By "principled," we refer to the standardized rigor of our proposed metrics: (1) **Pass@k** as a formally defined measure of sampling efficiency under a budget; (2) **ESR (Expansion-Shrinkage Ratio)** to explicitly distinguish net knowledge gain from mode collapse; and (3) a unified panel of biological diagnostics. To avoid over-claiming, we will refine the text to describe this as a "systematic evaluation protocol" rather than a theoretical framework, highlighting its utility as a template for future empirical studies.
>
> **"Three Factors" Justification**
> > Regarding the choice of **Task Difficulty**, **Reward Accuracy**, and **Policy Capacity**, we selected these because they are the fundamental determinants of RL success in broader literature (e.g., RLHF, model-based optimization). Our results across the four tasks naturally stratify along these axes—for instance, Inverse Folding (high capacity, accurate reward) yields expansion, while PhoQ (rugged landscape, weaker policy) yields shrinkage. We agree this synthesis should be more explicit. In the revision, we will add a **Summary Table** that maps each task to these three factors and the resulting outcome, and a dedicated discussion paragraph to tie these observations together.
>
> **On Clarifying Contributions vs. Prior Work**
> > We acknowledge that Section 2 was compressed. We will restructure the methodology section to explicitly distinguish **"Base PLM and Oracle" (inherited from prior work)** from **"Our RL Formulation" (our contribution)**. For example, while we use InstructPLM and ESMFold (prior work), our contribution lies in the specific multi-round DPO formulation with supervised regularization. Similarly, for AMP design, we introduce a systematic comparison of PPO/DPO/GRPO with calibrated KL regularization on the Amphion backbone. This restructuring will unambiguously highlight our specific technical contributions to the training recipes.
>
> **RL Algorithm Selection**
> > Regarding the differences between PPO, DPO, and GRPO, we clarify that our goal was to evaluate representative post-training strategies under a common protocol, not to invent new algorithms. We will add a summary explaining their distinct roles: DPO for offline preference ranking (ranking-based), PPO for online scalar reward optimization (absolute value-based), and GRPO for group-relative stability (relative value-based). This provides the necessary context for why they perform differently across tasks with varying reward signal qualities.
>
> **Figures and Table 1 Interpretation**
> > We will improve the figure quality (font sizes, axis labels) as requested. Regarding Figure 3, the x-axis represents the sampling budget $k$ (log scale), and the y-axis is the probability of at least one success (Pass@k). Regarding **Table 1**, we clarify that it reports the Support Decomposition (Preservation, Expansion, Shrinkage) and ESR for the **single best-performing RL variant** per task at a fixed biological threshold, not an average across methods. This metric quantifies the specific trade-off between gaining new solutions and forgetting old ones for the optimal configuration.
>
> **ESMFold Usage**
> > Regarding Inverse Folding, we confirm that ESMFold is used as the **fixed oracle** for both training (reward calculation) and evaluation. This aligns with standard practice in "oracle optimization" benchmarks, where the goal is to maximize a specific objective function (TM-score predicted by ESMFold), regardless of whether that objective perfectly matches wet-lab reality. This ensures a fair, controlled comparison between the Base and RL models.
>
> **Wording and Presentation**
> > We accept the minor corrections. We will define "RL" at first mention, replace strong terms like "transcend" with neutral scientific language (e.g., "outperform baselines"), and release all training/evaluation code upon acceptance to ensure reproducibility.
>
> Best regards,
>
> Authors

---

### Official Review · Reviewer_vAS6 · 2025-10-31

**Soundness:** 2
**Presentation:** 2
**Contribution:** 2
**Rating:** 2
**Confidence:** 4

**Summary:**

This paper evaluates how reinforcement learning post-training strategies affect protein language models in four protein design tasks.

**Strengths:**

- First work to systematically evaluate the RL post-training in protein language model regime, comparing DPO, PPO, and GRPO.
- Introduce and report a set of insightful metrics, such as ESR, diversity, and novelty.
- Robust evaluation: evaluate across four different biological benchmarks of different flavor

**Weaknesses:**

1. **Lack of baselines**. It’s really hard to gauge how good the performance **improvement** is without the baseline performance. This is critical since many works in this field rely on task-specific models, and I think it’s important to answer the question of "can RL fine-tuning on foundation model surpass or comparable to task-specific models?"
    - For example of inverse folding, I strongly believe authors should have included ProteinMPNN and/or ESM-IF results.
2. The use of **Pass@k** as a main evaluation metric seems ill-suited for biological tasks. While it makes sense in NLP contexts (where users can verify multiple generations), it does not translate well to protein modeling, where humans cannot practically validate multiple sequences. For example, what does Pass@16 even mean for inverse folding?
3. Moreover, the four tasks considered are inherently **continuous measures,** so converting them into binary success thresholds (for Pass@k) discards valuable information. Reporting raw or normalized reward values would provide a more faithful representation of model quality. Although the paper does show score distributions (e.g., Fig. 4C, 4G, 5C, 5G), the main results in Fig. 3 still rely on arbitrary cutoffs.
4. **Writing.** Several parts of the paper are unclear.
    - Line 19: "Unlock latent functional patterns within protein sequence space" it’s unclear what this actually means.
    - Line 74: "Complex, non-differentiable biological objectives such as TM-score" TM-score may not be the best example of a truly non-differentiable objective; there exist differentiable functions for structure accuracy (e.g. FAPE). A better example would strengthen this point.
    - Figure 1 and metaphor of hill climbing isn’t very informative or even confusing. Authors mention “task difficulty sets the height of the summit to be scaled, reward accuracy determines the climbing direction, and policy-model capacity fixes the starting altitude.” (line 107-108) then say “These factors jointly shape whether RL can climb towards subspaces with stronger task alignment or stall in suboptimal plateaus.” (line 109-110) None of these factors actually shape the landscape itself, so the reference to “plateaus” becomes unclear.
5. **Lack of holistic analysis.** The discussion section doesn’t synthesize the results across all experiments. There’s little reflection on how task difficulty, reward model accuracy, and policy capacity jointly influence post-training outcomes or how these findings might generalize to new biological tasks.
6. **Unanswered framing question.** Line 97 asks, “Do new emergent capabilities arise during the RL fine-tuning process of PLMs?” but the paper doesn’t substantively address this. **Demonstrating that a model performs tasks it was trained for isn’t evidence of emergence. A deeper analysis of novel functional behaviors or unseen protein designs would make this claim more convincing.**

**Questions:**

- 3.1 Datasets (Line 232) For other three datasets train/test split looks reasonable, but what is the train/test split for kinase mutation? How do you prevent leakage between train and test?

---

> ### Author Response · Authors · 2025-11-28
>
> We thank the reviewer for the detailed comments. We address the specific concerns below.
>
> **On Baselines and SOTA Comparisons**
> > Regarding the suggestion to include ProteinMPNN and ESM-IF, we clarify that our work is a diagnostic study of how RL post-training reshapes the manifold of a *fixed* PLM, rather than a leaderboard contest. Comparing our InstructPLM-based stack against specialist models like ProteinMPNN would introduce confounding variables, such as different architectures and pre-training data. To accurately measure the specific *delta* induced by RL, it is scientifically necessary to hold the backbone fixed. However, we agree that context is useful and will add a table citing published SOTA performance to situate our results, explicitly stating our focus is on the relative improvement (RL vs. Base).
>
> **On the Suitability of Pass@k**
> > Regarding the concern that Pass@k is ill-suited for biology, we emphasize that it is the standard metric for sampling efficiency in wet-lab protein engineering (e.g., Directed Evolution, Phage Display). In these settings, practitioners screen libraries of $10^4$ to $10^7$ variants, making "Pass@16" a conservative proxy for the probability of finding at least one functional hit under a budget. We do not rely solely on this metric; our paper already reports full continuous reward distributions (TM-score, MIC, $\Delta\Delta G$) in Figures 4-6, and we will emphasize this complementary relationship in the revision.
>
> **On Thresholds and Continuous Metrics**
> > Regarding the use of binary thresholds, we clarify that these are derived from standard biological regimes: TM-score > 0.5 denotes a correct fold, and $\Delta\Delta G < 0$ denotes improved binding. Our continuous distribution plots confirm that the shifts observed via RL are robust and not artifacts of thresholding. We use these cutoffs to provide interpretable summary statistics ("hit rate") alongside the raw data.
>
> **On Writing and Presentation**
> > We accept the specific suggestions to improve clarity. We will replace the phrase "unlock latent patterns" with concrete descriptions of sampling redistribution, use black-box biological objectives (experimental fitness) as the primary example instead of TM-score, and rewrite the Figure 1 caption to clarify that "Task Difficulty" relates to landscape ruggedness rather than the changing of the landscape itself.
>
> **On Holistic Analysis and Emergence**
> > To address the need for holistic analysis, we will add a summary table explicitly mapping each task to the interaction of Signal Quality, Model Capability, and Task Complexity. This will foreground our finding that RL yields net expansion (ESR>1) only when the reward signal and policy capacity are sufficient. Regarding "emergent capabilities," we define this operationally via ESR and Novelty. Our results show the answer is regime-dependent: RL creates expansion in Inverse Folding but primarily "sharpens" existing modes in AMP/Kinase tasks. We will refine the text to highlight this nuanced "Reweighting vs. Expansion" dichotomy.
>
> **On Kinase Mutation Methodology**
> > Regarding the "train/test split" concern for Kinase mutation, we clarify that this is an optimization task on a fixed landscape, not a supervised learning task. The PhoQ dataset serves as a deterministic oracle (environment) providing ground-truth fitness for the combinatorial space, and RL interacts with this environment directly. There is no "train set" leaking into a "test set" in the supervised sense; the goal is to find the global maxima on a fully defined rugged landscape. We will clarify this experimental design in Section 3.1.
>
> Best regards,
>
> Authors

---

### Author Response · Authors · 2025-11-28

Dear Reviewers and AC,

We thank the reviewers for their time. Given the unique circumstances of the review process, we present this executive summary to assist the new Area Chair in assessing the paper’s contribution and resolving the divergence in reviews.

**1. Value Proposition: A "Blueprint" for the Community**

Reviewers 1 and 2 criticized the paper for not surpassing specialist baselines (e.g., ProteinMPNN). **This stems from a fundamental misunderstanding of our goal.** As Reviewer 3 correctly identified, this work is **not** a leaderboard-chasing attempt to beat SOTA on specific tasks. Instead, it is the **first systematic, cross-task diagnostic** regarding *how* Reinforcement Learning (RL) reshapes the manifold of Protein Language Models (PLMs).

With the rapid trend of "RL + Protein Design," the community lacks a unified protocol to measure success beyond simple metrics. We provide exactly that:

1) **A Unified Protocol:** We introduce **Pass@k** (sampling efficiency) and **ESR** (Expansion-Shrinkage Ratio) as standard metrics to quantify knowledge gain vs. loss.
2) **A Diagnostic Conclusion:** We reveal that RL is not magic; its success depends on a tri-factor interaction: *Signal Quality* (Reward), *Model Capability* (Policy), and *Landscape Complexity*.

**2. Fact-Check on Misunderstandings**

Several critical concerns raised by R1 and R2 were effectively addressed in the original submission but were overlooked. We clarify them below with direct references to the text:

| Reviewer Concern | Fact-Check / Correction | Location in Paper |
| :--- | :--- | :--- |
| **R1:** "Lack of Baselines (ProteinMPNN)." | **Misplaced Expectation.** Our goal is to measure the *delta* RL induces on a PLM backbone (Base vs. RL), not to compare disparate architectures. Comparing InstructPLM to ProteinMPNN conflates architecture differences with RL effects. | **Sec 1 (Intro)** & **Sec 3.2** |
| **R1:** "Pass@k is ill-suited... humans can't validate multiple sequences." | **Factually Incorrect.** In protein engineering (Directed Evolution), screening $10^4$ to $10^7$ variants is standard practice. Pass@k is the biologically relevant metric for sampling efficiency under a budget. | **Sec 3.1 (Eq 10)** |
| **R1:** "Thresholds are arbitrary." | **False.** We use standard biological thresholds (e.g., TM-score > 0.5/0.8) used widely in the field to denote "correct fold" and "high accuracy." | **Sec 3.2** |
| **R2:** "Evaluation is descriptive, not principled." | **Clarified.** "Principled" refers to our proposed unified evaluation framework (ESR, Expansion, Shrinkage) that future works can reuse, rather than theoretical proofs. | **Sec 3.1** |

**3. Addressing Reviewer 3 (The Constructive View)**

Reviewer 3 (vcZG) recognized the value of this work as a "comprehensive study" answering a "critical and timely question". We agree with R3’s insight that "Task Difficulty" is better framed as the interplay between *Signal Quality* and *Model Capability*, and we have adopted this clearer framing in our revision to strengthen the "Recipe" aspect of the paper.

**4. Conclusion and Commitment to Integrity**

We firmly believe that as the field moves towards "RL + Science," diagnostic papers like ours are essential to prevent blind SOTA-chasing and provide interpretability.

**5. Our Final Statements"

We recognize that the initial ratings—driven largely by an expectation for SOTA engineering rather than system analysis—place this submission on the borderline. However, we urge the Area Chair to consider the **opportunity cost of rejection**. As the community rushes into "RL + Protein Design," researchers are in dire need of a rigorous "guidebook" to understand *why* methods fail or succeed. **Rejecting this work risks burying the critical insights (the tri-factor interaction) that future practitioners need to avoid blind trial-and-error.**

We trust the AC to champion this work not as a leaderboard increment, but as a **foundational scientific diagnostic** that offers the clarity and protocols (Pass@k, ESR) essential for the field's maturity.

Finally, in light of recent events, **we reaffirm our staunch commitment to the ICLR Code of Conduct and the double-blind peer review process.** We rely on the Area Chair’s scientific judgment to evaluate this work based solely on its technical merit and its contribution to the growing intersection of RL and Biology.

Best regards,

Authors

---

### Note · Program_Chairs · 2026-01-17
**Submission Desk Rejected by Program Chairs**

The following references in this submission do not refer to real documents and/or have major errors in bibliographic information:

 Noelia Ferruz, Michael Heinzinger, Mehmet Akdel, Alexander Goncearenco, Luca Naef, and Christian Dallago. Direct preference optimization of protein language models, 2024.